# Development and Validation of a Risk Prediction Tool to Identify People at Greater Risk of Having Hepatitis C among Drug Users

**DOI:** 10.3390/ijerph192315677

**Published:** 2022-11-25

**Authors:** Gang Huang, Wei Cheng, Yun Xu, Jiezhe Yang, Jun Jiang, Xiaohong Pan, Xin Zhou, Jianmin Jiang, Chengliang Chai

**Affiliations:** 1Medical School of Ningbo University, Ningbo University, Ningbo 315211, China; 2Department of AIDS and STDs Prevention and Control, Zhejiang Province Center for Disease Control and Prevention, Hangzhou 310051, China; 3Key Lab of Vaccine, Prevention and Control of Infectious Disease of Zhejiang Province, Hangzhou 310051, China

**Keywords:** drug use, HCV prevalence, risk behaviors, risk score

## Abstract

Background: People who use drugs (PWUD) are among those with the highest risk for hepatitis C virus (HCV) infection. Highly effective direct-acting antiviral agents offer an opportunity to eliminate HCV. A simple tool for the prediction of HCV infection risk in PWUD is urgently needed. This study aimed to develop and validate a risk prediction tool to identify people at greater risk of having hepatitis C among PWUD that is applicable in resource-limited settings. Methods: We extracted data from national HIV/AIDS sentinel surveillance in PWUD (Zhejiang Province, 2016–2021) and developed and validated a risk score to improve HCV testing in PWUD. This risk score consists of seven risk factors identified using multivariable logistic regression modeling (2016–2020, exploratory group). We validated this score using surveillance data for 2021 (validation group). The accuracy of the model was determined using C-statistics. Results: We identified seven risk factors, including sex, age, marital status, educational attainment, and the use of heroin, morphine, and methamphetamine. In the exploratory group, the positive rates of detecting the HCV antibody in the low-risk (0–9 points), intermediate-risk (10–16 points), and high-risk (≥17 points) groups were 6.72%, 17.24%, and 38.02%, respectively (*P_trend_* < 0.001). In the validation group, the positive rates in the low-, medium-, and high-risk groups were 4.46%, 12.23%, and 38.99%, respectively (*P_trend_* < 0.001). Conclusions: We developed and validated a drug-specific risk prediction tool for identifying PWUD at increased risk of HCV infection. This tool can complement and integrate the screening strategy for the purpose of early diagnosis and treatment.

## 1. Introduction

Hepatitis C virus (HCV) infection causes a significant mortality and morbidity burden, particularly in developing countries. According to data from the World Health Organization (WHO), there were approximately 58 million people with chronic HCV infection in 2019, and approximately 1.5 million people are newly infected with HCV every year. The annual death toll associated with HCV is 290,000, which is unacceptably high [1]. Most of these deaths are caused by chronic liver disease and liver cancer [2]. HCV-associated mortality and morbidity can be meaningfully reduced by implementing prevention, screening, and treatment interventions [3].

Direct-acting antiviral agents (DAAs) are highly effective medications that have turned HCV infection into a curable disease [4]. Hence, the WHO is aiming for HCV elimination by the year 2030. To this end, the WHO issued a report estimating that 90% of infected individuals need to be diagnosed and 80% need to be treated in order to achieve this goal [1]. Myriad challenges remain, as many infected individuals are unaware of their infection status, only 20% of the world’s 71 million HCV cases are diagnosed, and the rate of HCV treatment uptake is <10% [5]. Therefore, the first essential step of accurately and consistently identifying HCV infection has become a focus of public health programming—especially programming implemented among those at high risk of HCV infection.

People who use drugs (PWUD) are the major drivers of HCV infection because of their low rates of screening, treatment, or treatment acceptance, along with the persistent risk of reinfection in this vulnerable subgroup [6]. Drug use is a major public health problem, since it is often associated with risky sexual behavior and blood-contact behavior, such as sharing needles and other injection equipment [7,8]. As of the end of 2020, the global population of those suffering from drug addiction had reached 270 million, including 1.801 million affected individuals in China [9]. Moreover, the rapidly growing prevalence of drug use and associated behaviors has led to concentrated epidemics of HIV and viral hepatitis among PWUD in many countries [10]. Concurrently, there has been an increase in the variety of new synthetic drugs (e.g., ATS, ketamine) available on the market, and an increasing number of health problems are observed in users of these new drugs [11].

Several screening strategies have been recommended to the general population and to high-risk groups. According to the US Preventive Services Task Force (USPSTF), one-off HCV screening is recommended for all people aged ≥18 years, and annual HCV testing is recommended for all high-risk groups [12]. Another study reviewed different virological tests using serum or plasma taken via venipuncture, including point-of-care tests, dried blood spots, and nucleic acid testing [13]. However, low screening rates and ongoing risk of reinfection were still barriers to achieving the elimination targets—especially among marginalized populations such as PWUD [14,15]. Improved HCV screening strategies, such as active and regular testing of those at greater risk of HCV infection, are essential to ensure timely diagnosis, linkage to care, and treatment initiation.

Therefore, this study aimed to developed a simple risk score that is applicable in resource-limited settings to predict those at greater risk of HCV infection among drug users. Our risk prediction tool can complement current screening strategies for further targeting those most vulnerable to HCV infection.

## 2. Materials and Methods

### 2.1. Subject Recruitment and Study Design

The national HIV sentinel surveillance (NHSS) program in China conducts annual serial and anonymous cross-sectional surveys among PWUD. The surveys are administered from April to June. The data for this study were extracted from the sentinel surveillance in Zhejiang Province (2016–2021); this sentinel surveillance effort is a part of the NHSS. The target sample size for each sentinel site was approximately 400 individuals, which included patients at eight compulsory drug rehabilitation centers in five cities in Zhejiang Province. Compulsory drug rehabilitation centers, which are under the jurisdiction of the Ministry of Justice, enroll individuals who have been caught using drugs and evaluated as having a substance use disorder. All PWUD received physical rehabilitation and psychological treatment in compulsory drug rehabilitation centers, including methadone treatment. The participants in compulsory drug rehabilitation centers were recruited using consecutive sampling. Each year, we conducted a survey of PWUD entering compulsory drug rehabilitation centers. To ensure data quality, we assigned a staff member who did not perform interviews to examine the questionnaires for errors, blanks, or inconsistent data.

### 2.2. Data Collection

Each study subject was interviewed face-to-face using a structured questionnaire, and interviews were conducted by an experienced field epidemiology investigator. From 2016 to 2021, a total of 20,877 interviews were conducted. The questionnaire collected information on demographic characteristics, drugs used in the past month, and sexual activity.

Demographic characteristics included sex, age, marital status, registered residence, ethnicity, and educational level. We also collected information on the use of 10 types of drugs: heroin, cocaine, opium, cannabis sativa, morphine, methamphetamine, demerol, ketamine, ecstasy, and magu (an amphetamine-type stimulant whose main ingredients are methamphetamine and caffeine). Regarding sexual activity, we asked participants about having sex after taking drugs in the past year.

Blood samples were collected for HCV testing by professional laboratory personnel. Antibody screening for HCV was performed via enzyme-linked immunosorbent assay (ELISA). Samples that were initially positive on the first ELISA screening were retested using another ELISA method. If both screening tests were positive, the sample was considered to be HCV-positive. The sample was considered to be negative if there was no positive reaction.

### 2.3. Statistical Analysis

All data were entered into the NHSS using client software. Categorical variables were expressed using frequencies (percentages). The study’s subjects were divided into two groups: those identified from the 2016–2020 surveillance data (the exploratory group), and those identified from the 2021 surveillance data (the validation group).

For the exploratory group, a binary logistic regression model was used to evaluate risk factors and calculate odds ratios (ORs). Variables suggestive of an association with HCV antibody positivity (*p* < 0.1) were considered to be candidate variables in multivariable model building. The alpha level was set at 0.05 in the final multivariable model.

We further developed our predictive risk score based on tools reported in previous studies [16,17,18]. After determining the final regression model, the scores for each predictor were assigned; these scores were proportional to the β regression coefficients. A risk score was then calculated for each subject. The variable with the smallest beta coefficient was considered to have a risk score of 1. We divided each beta coefficient by the smallest beta coefficient and rounded to the nearest whole number to determine the score for each predictor variable. The study’s subjects were categorized into three groups (by setting cutoff points at the 25th and 75th percentiles of the model’s total risk score distribution) based on their subsequent risk of HCV infection (i.e., low-, medium-, or high-risk) [17].

Participants identified from the 2021 surveillance data were used as the validation group to assess the risk score model. The risk scores of variables in the exploratory group were applied in the validation group. We calculated the total points for each subject and classified the low-, medium-, and high-risk groups as described above. To test the performance of the scoring tool, we compared the rates of positive HCV antibody status among PWUD in the low-, medium-, and high-risk groups and calculated C-statistics for each model. We also compared the proportional difference in predicted HCV antibody positivity between the high- and low-risk groups. Cochran–Armitage tests were used to determine trends in the risk of HCV infection with reference to an increasing number of points and classification categories.

Data were cleaned and organized using Excel 2016 software (Microsoft Corporation, Seattle, WA, USA). R statistical software (v.4.1.2, The R Project for Statistical Computing) was used for statistical analysis.

### 2.4. Ethics

The described HIV sentinel surveillance system is a nationwide governmental public health program. Participants were not required to provide personal identifiers (e.g., information on their name, address, and occupation) as a condition for enrollment. Verbal consent was obtained from all participants prior to the survey administration and specimen collection. This work was conducted in accordance with the principles of the Declaration of Helsinki.

All of the surveillance work was completed by the local Centers for Disease Control and Prevention (CDC). We extracted data from the provincial HIV sentinel surveillance system. We did not collect extra information or specimens for this study. Therefore, our study is based on data derived from routine work.

## 3. Results

From 2016 to 2020, a total of 17,871 PWUD were enrolled in the exploratory group, with an average age of 37.5 years. Approximately 75.75% of the drug users were aged >30 years, 86.39% were male, 61.69% were married, and 67.61% had a secondary-school-level educational attainment. Among the 10 drug types evaluated herein, methamphetamine and heroin were the most commonly used (68.92% and 29.99%, respectively). In the bivariate analysis, sex, age, marital status, education level, and the use of heroin, morphine, ketamine, methamphetamine, and magu were each associated with HCV infection (*p* < 0.1; Table 1).

In the multivariate analysis, various factors (including female sex, marriage, low educational attainment, advanced age, and heroin and methamphetamine use) continued to predict HCV antibody positivity. Compared to those aged ≤30 years, PWUD between the ages of 30 and 40 years (adjusted OR (aOR) = 2.28, 95% confidence interval (CI): 2.00–2.60) and those aged >40 years (aOR = 3.60, 95% CI: 3.14–4.12) showed increased HCV antibody positivity. Compared to those with a college education or above, PWUD with illiteracy (aOR = 1.51, 95% CI: 1.12–2.04), a primary education level (aOR = 1.59, 95% CI: 1.20–2.10), a secondary education level (aOR = 1.56, 95% CI: 1.19–2.05), or a high school/technical secondary education level (aOR = 1.40, 95% CI: 1.05–1.87) showed higher rates of HCV antibody positivity. Positive HCV antibody status was also associated with sex (aOR = 1.48, 95% CI: 1.32–1.65), marital status (married: aOR = 1.41, 95% CI: 1.39–1.54), heroin use (aOR = 4.50, 95% CI: 3.88–5.22), morphine use (OR = 1.71, 95% CI: 1.14–2.59), and methamphetamine use (aOR = 1.17, 95% CI: 1.01–1.36) (Table 2).

We assigned risk scores for each predictor based on β regression coefficients. The highest and lowest risk scores of 9 and 1 points occurred in heroin and methamphetamine users, respectively (Table 2). Moreover, to assess the risk of HCV antibody positivity in PWUD according to participant characteristics, we calculated total risk scores for each PWUD. Participants had total risk scores ranging from 0 to 25. Based on the percentile distribution of total risk scores, participants were divided into three groups: low-risk (0–9 points), medium-risk (10–16 points), and high-risk (≥17 points). Most HCV antibody positivity (1694/4455) occurred in the high-risk group. The rates of positive HCV antibody status were 6.72%, 17.24%, and 38.02% in the low-, medium-, and high-risk groups, respectively (*P_trend_* < 0.001) (Figure 1A), and the simplified integer-based risk score performed well in the exploratory group (C-statistic = 0.71) (Figure 2A). The differential probability of HCV antibody positivity between the high- and low-risk groups was 0.31.

A total of 3006 PWUD were included in the validation group. Of these, 462 (15.37%) PWUD were positive for HCV. When the developed risk score was applied, the positivity rates increased with increasing risk scores (4.46%, 12.23%, and 38.99% in the low-, medium-, and high-risk groups, respectively; *P_trend_* < 0.001) (Figure 1B). The C-statistic was 0.75, and the differential probability of HCV antibody positivity between the high- and low-risk groups was 0.35 (Figure 2B).

## 4. Discussion

PWUD are among the groups at highest risk of HCV infection. Timely screening, identification of HCV infection among PWUD, and provision of treatment are of urgent priority to achieve the WHO’s international strategy and action plan of eliminating any impact of HCV by 2030. Since no previous studies have attempted to develop a hepatitis C risk prediction model for PWUD in China, the purpose of our study was to develop a simple scoring system to predict the risk of HCV infection in PWUD based on drug type. In the exploratory group enrolled in the present study, the positivity rates for the HCV antibody in the low-, medium-, and high-risk groups were 6.72%, 17.24%, and 38.02%, respectively (C-statistic = 0.71). Moreover, on validation, the positivity rates for the HCV antibody in the low-, medium-, and high-risk groups were 4.46%, 12.23%, and 38.99%, respectively (C-statistic = 0.75). Therefore, we conclude that this scoring system can easily identify PWUD at greater risk of HCV infection and can be applied as a complementary tool to enhance screening strategies for HCV infection in PWUD.

The results of the multivariate models revealed that HCV antibody positivity was associated with demographic characteristics, including sex, older age, marital status, and educational attainment, consistent with the results of many previous studies [19,20,21,22]. Moreover, studies have shown that the higher rate of HCV antibody positivity in women seen across studies can be attributed to differences in biological and behavioral factors, and that women who inject drugs have a higher prevalence of HCV-related risk exposures than men [21]. Older age was associated with HCV infection in PWUD in the present study, which may have been related to the duration of drug use [20,22]. More specifically, it is reasonable to assume that the longer the period of drug use, the greater the chance of exposure to HCV contamination. In this study, being married was independently associated with infection, and the HCV positivity rate of married people was 1.41 times higher than that of unmarried people. This association was also observed in various international studies, and this trend may be related to intra-family transmission as well as having multiple sexual partners [23,24]. In addition, we found that the lower the education level, the higher the HCV positivity rate. Multiple studies have found that lower education levels are associated with poor self-protection awareness [25,26]. Additionally, the educational level of older people tends to be lower. Therefore, screening and treatment are essential in this potentially vulnerable subgroup.

Although research shows that drug use is common before or during sexual activity [27], we did not find an association between sexual activity after taking drugs and HCV infection. Several studies have indicated that sexual transmission of HCV is an inefficient and rare mode of transmission, especially among HIV-negative people [28,29]. Therefore, we conclude that the construction of our prediction model based on drug use is reasonable and efficient.

Our study showed that 30.0% of PWUD reported using heroin and 68.9% of PWUD reported using methamphetamine; this is similar to the surveillance findings reported in other provinces in China [30,31]. According to the Drug Situation Report in China in 2020, there were 1.801 million drug addicts in China in 2020, and methamphetamine and other abused species still maintained a relatively large scale [9]. It should be noted that methamphetamine is the most abused drug in China. The aORs for positive HCV antibody status with respect to the use of heroin, morphine, and methamphetamine were 4.50, 1.71, and 1.17, respectively. These results are also consistent with the findings of a study conducted in Jiangsu Province [30].

It should be noted that heroin users are more likely to be infected with HCV because they are more likely to administer the drug by intravenous injection and are more likely to share needles [32]. A multistage systematic review estimated that 82.9% (76.6%–88.9%) of persons who inject drugs (PWID) primarily injected opioids [33]. In contrast, Moradi et al. found that smoking using a pipe was the most common method of methamphetamine use, which has a lower associated risk of HCV infection than injection [34]. Heroin use showed the highest risk score; therefore, regular follow-up and periodic HCV testing for heroin users should be given priority, despite the decreased incidence of heroin use among PWUD.

Regression-based methodologies encompassing a wide range of risk factors have determined potential associations between evaluated risk factors and HCV infection, and could be promising tools for improving HCV testing strategies in various settings. For example, Newsum et al. developed and validated the HCV-MOSAIC risk score, which consists of six self-reported risk factors identified using multivariable logistic regression [18]. Using this risk score, if 42–59% of HIV-infected men who have sex with men (MSM) undergo HCV testing, 73–100% of HIV-infected MSM could be correctly identified as having acute HCV infection. Javier et al. also created a score based on five items identified through logistic regression modeling in order to predict the risk of HCV infection among the general population, achieving an area under the curve (AUC) of 0.896 (95% CI: 0.833–0.959) [16]. To the best of our knowledge, no previous study has attempted to develop a risk score predicting HCV infection among PWUD based on drug type. Therefore, our study complements prior studies directed at improving screening for HCV infection. According to the findings of the present research, we suggest that each PWUD should complete a questionnaire collecting information on sex, age, marital status, education, and the use of heroin, morphine, and methamphetamine when sent to a compulsory drug rehabilitation center. After preliminary evaluation, for the medium- and high-risk groups, intensified intervention and screening—including harm reduction services, health education, active and repeated HCV testing, and pre- and post-test counseling—should be offered in a timely manner. For the low-risk subjects, one-off testing should be 100% covered. Moreover, since our risk prediction tool is easy to implement, we also encourage drug users to adopt this tool to perform self-evaluation and, thus, facilitate active HCV testing.

Despite the significant advantages of this simple risk prediction scoring model, this study has some limitations that we must acknowledge herein. First, our study was cross-sectional and, therefore, could not determine a causal relationship between the investigated factors and HCV antibody positivity. Second, we only collected information on drugs used in the past month. Therefore, we could not discriminate the effects of previous drug use on the risk of HCV infection. However, investigating recent drug history could reduce recall bias. Third, our study developed an informative risk score based on drug type, and we did not include variables such as injection history or sharing needles in the model, because of the lack of information for each type of drug. However, the different aORs of each drug for HCV antibody positivity indicate that different types of drugs are linked with different risky behaviors. For example, studies have found that heroin users are more likely to constitute a higher proportion of those with a history of injection or sharing needles [32]. Therefore, we should improve our questionnaire to collect as much information as possible about the use patterns of each drug and assess their impacts on our prediction model. Fourth, several factors were not investigated in this study, such as history of incarceration, duration of drug use, tattoos, and sharing of razors or other personal items while incarcerated, which also could be considered to be risk factors in HCV infection. The impacts of these factors should be evaluated via prediction models in future studies [20,35]. Finally, since our study’s subjects were all PWUD enrolled in rehabilitation centers in Zhejiang Province, our findings should be extrapolated with caution—especially with regard to application within community populations. We plan to further validate the performance of our risk score across different settings in future research efforts.

## 5. Conclusions

Herein, we developed and validated a simple risk prediction tool for identifying people at greater risk of HCV infection among PWUD. The identified predictors of HCV infection included sex, marital status, education, age, and the use of heroin, morphine, and methamphetamine. Although our findings require additional validation, we believe that this easy-to-implement risk prediction tool described herein will complement and improve HCV screening and management strategies in compulsory drug rehabilitation centers. By targeting higher-risk patients, we can effectively encourage PWUD who may be underutilizing treatment to engage in effective HCV prevention, intervention, and treatment services.

## Figures and Tables

**Figure 1 ijerph-19-15677-f001:**
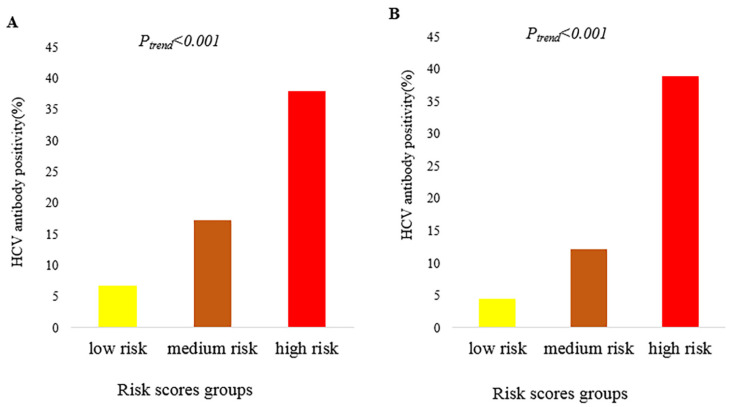
The HCV antibody positivity rates of PWUD in the low-, medium-, and high-risk groups: (**A**) exploratory group; (**B**) validation group.

**Figure 2 ijerph-19-15677-f002:**
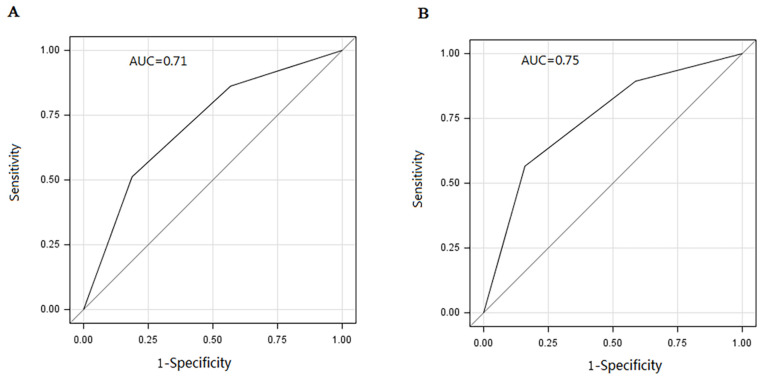
ROC curves for HCV antibody positivity risk score groups: (**A**) exploratory group; (**B**) validation group.

**Table 1 ijerph-19-15677-t001:** Demographic characteristics and drugs used among PWUD, by HCV antibody status (exploratory group, 2016–2020).

Characteristics	Total (n = 17,871)	Positive HCV Antibody Status, n (%)	Negative HCV Antibody Status, n (%)	Unadjusted OR (95% CI)	*p*-Value
Sex					<0.001
Female	2432	511 (21.0)	1921 (79.0)	1.21 (1.09–1.34)	
Male	15,439	2788 (18.1)	12,651 (81.9)		
Household registration					0.610
Zhejiang Province	11,638	2161 (18.6)	9477 (81.4)	0.98 (0.91–1.06)	
Other provinces	6233	1138 (18.3)	5095 (81.7)		
Age (years)					<0.001
≤30	4333	366 (8.4)	3967 (91.6)		
31–40	6994	1180 (16.9)	5814 (83.1)	2.20 (1.94–2.49)	
≥41	6544	1753 (26.8)	4791 (73.2)	3.97 (3.52–4.47)	
Marital status					0.013
Married/cohabiting	11,025	2098 (19.0)	8927 (81.0)	1.11 (1.02–1.20)	
Divorced/single	6846	1201 (17.5)	5645 (82.5)		
Education					<0.001
Illiteracy	1343	329 (24.5)	1014 (75.5)	2.79 (2.10–3.70)	
Primary school	4446	928 (20.9)	3518 (79.1)	2.27 (1.74–2.96)	
Middle school	8819	1553 (17.6)	7266 (82.4)	1.84 (1.41–2.38)	
High school/technical secondary school	2630	423 (16.1)	2207 (83.9)	1.65 (1.25–2.17)	
College degree or above	633	66 (10.4)	567 (89.6)		
Sexual activity after taking drugs					0.914
Yes	7122	1312 (18.4)	5810 (81.6)	1.00 (0.92–1.08)	
No	10,749	1987 (18.5)	8764 (81.5)		
Heroin use					<0.001
Yes	5359	1922 (35.9)	3437 (64.1)	4.52 (4.18–4.89)	
No	12,512	1377 (11.0)	11,135 (89.0)		
Cocaine use					0.215
Yes	75	18 (24.0)	57 (76.0)	1.40 (0.82–2.38)	
No	17,796	3281 (18.4)	14,515 (81.6)		
Opium use					0.624
Yes	33	5 (15.2)	28 (84.8)	0.79 (0.30–2.04)	
No	17,838	3294 (18.5)	14,544 (81.5)		
Cannabis sativa use					0.157
Yes	150	21 (14.0)	129 (86.0)	0.72 (0.45–1.14)	
No	17,721	3278 (18.5)	14,443 (81.5)		
Morphine use					0.023
Yes	139	36 (25.9)	103 (74.1)	1.55 (1.06–2.27)	
No	17,732	3263 (18.4)	14,469 (81.6)		
Methamphetamine use					<0.001
Yes	12,316	1514 (12.3)	10,802 (87.7)	0.30 (0.27–0.32)	
No	5555	1785 (32.1)	3770 (67.9)		
Demerol					0.256
Yes	172	26 (15.1)	146 (84.9)	0.79 (0.52–1.19)	
No	17,699	3273 (18.5)	14,426 (81.5)		
Ketamine use					0.058
Yes	240	33 (13.8)	207 (86.2)	0.70 (0.49–1.01)	
No	17,631	3266 (18.5)	14,365 (81.5)		
Ecstasy use					0.112
Yes	98	12 (12.2)	86 (87.8)	0.62 (0.34–1.13)	
No	17,773	3287 (18.5)	14,486 (81.5)		
Magu use					<0.001
Yes	556	59 (10.6)	497 (89.4)	0.52 (0.39–0.68)	
No	17,315	3240 (18.7)	14,075 (81.3)		

CI, confidence interval; HCV, hepatitis C virus; OR, odds ratio; PWUD, people who use drugs.

**Table 2 ijerph-19-15677-t002:** Multivariable logistic regression analysis and derivation of the risk score classification in the exploratory group (2016–2020).

Variables in the Final Model	*B*	*p-Value*	*Adjusted OR*	*95% CI*	*Points*
Sex (compared to males)	0.39	<0.001	1.48	1.32–1.65	2
Marital status (compared to divorced/single)					
Married	0.34	<0.001	1.41	1.29–1.54	2
Education (compared to college degree or above)					
Illiteracy	0.41	0.007	1.51	1.12–2.04	3
Primary school	0.46	0.001	1.59	1.20–2.10	3
Middle school	0.45	0.001	1.56	1.19–2.05	3
High school or technical secondary school	0.34	0.022	1.4	1.05–1.87	2
Heroin use	1.5	<0.001	4.5	3.88–5.22	9
Morphine use	0.54	0.011	1.71	1.14–2.59	3
Methamphetamine use	0.16	0.039	1.17	1.01–1.36	1
Age group (compared to ≤30 years)					
31–40	0.82	<0.001	2.28	2.00–2.60	5
>40	1.28	<0.001	3.6	3.14–4.12	8

CI, confidence interval; OR, odds ratio.

## Data Availability

The data presented in this study are available on request from the corresponding author. Due to the data design privacy information, these data are not publicly available.

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
