# Peer review of "Development and Validation of a Risk Prediction Tool to Identify People at Greater Risk of Having Hepatitis C among Drug Users"

_ijerph, 2022, doi:10.3390/ijerph192315677_

Round 1
Reviewer 1 Report
Major comment:
Dear authors,
from a methodological and scientific point of view I have almost nothing to object to your study. The objectives are clear, the introduction is complete, the methodology is solid, the results are shown correctly.
The only huge concern is about the conclusions, and in general the purpose of the article. In fact, I believe that suggesting the use of a risk evaluation tool for hepatitis C, in a high-risk group such as PWUD, is wrong from the point of view of Public Health policy. In fact, although your tool is functional, this does not mean that it is useful and appropriate to use it.
In your tool, among the low-risk subjects, a prevalence between 4% and 6% emerges, in any case a considerably high prevalence, and worthy of complete screening. The WHO and other relevant national and international Public Health organizations suggest screening on at least an annual basis in the high-risk groups. The WHO never suggests the use of risk prediction tools among high-risk subjects. Similarly, the Systematic Review you cite [14] does not refer to these strategies among PWUDs and other high-risk groups, but includes the use of risk scores among the general population, to avoid universal screening whenever possible.
In the discussion (lines 276-281) you hypothesize the use of the risk score as a step to decide who should be tested and who should not. This is, in my opinion, an inadequate suggestion in this category of patients. Your risk prediction tool would be very effective if applied to a low-risk population, to optimize screening for HCV-Ab, but it is in my opinion totally inadequate in a high-risk population, where screening must be conducted in all subjects, without risk stratification.
I suggest changing the core of the paper, presenting the score not as a replacement strategy to screening, but as an integrative strategy: this score could be used to change the approach in screening (for example, active and repeated offer in subjects at risk high and moderate, passive and one-off offer in low risk subjects), or it could be used as a tool to “motivate” PWUDs to undergo the test more frequently, showing them evidence of their risk. In all cases, this tool must complement and integrate the screening test proposal on an annual basis with PWUDs, and never replace it.
Minor comment: The title is misleading. The tool is not used to identify PWUDs with Hepatitis C, but those at greater risk of having Hepatitis C.
Reviewer 2 Report
2.1. Subject recruitment and study design
The authors must detail the procedure to collect the data, in correspondence with the number of data analyzed (n = 17,871)
2.2. Data collection
Indicate the number of interviews conducted.
3. Results
Check the values in table 1, some do not match. In the manuscript, data that do not match appear in highlight
